# Emergency Animal Boarding: A Social Return on Investment

**DOI:** 10.3390/ani13142264

**Published:** 2023-07-10

**Authors:** Gemma C. Ma, Jioji Ravulo, Ursula McGeown

**Affiliations:** 1Royal Society for the Prevention of Cruelty to Animals New South Wales, Yagoona, NSW 2199, Australia; umcgeown@rspcansw.org.au; 2Sydney School of Veterinary Science, The University of Sydney, Camperdown, NSW 2006, Australia; 3Sydney School of Education and Social Work, The University of Sydney, Camperdown, NSW 2006, Australia; jioji.ravulo@sydney.edu.au

**Keywords:** companion animal, dog, cat, human–animal bond, social return on investment, animal shelter, animal welfare

## Abstract

**Simple Summary:**

Companion animals are valued members of many families, providing companionship, affection, and someone to nurture and love. Some situations such as being suddenly hospitalized for mental or physical health or becoming homeless make it difficult for people to keep their animal companions, especially for those who have a limited support network. The RSPCA NSW Emergency Boarding and Homelessness program supports people experiencing a crisis to access boarding and veterinary treatment for their animals, helping to keep the family together in the long term. This study aimed to understand the social value created by the program and to express this value in monetary terms. We interviewed 13 stakeholders including program clients, RSPCA Inspectors, and human service providers to understand what changed for them because of the program and how they valued that change. The main changes were experienced by program clients and their animals. The most valuable changes were being able to keep their companion animal and improved mental health and wellbeing. We estimate that this program results in social value worth AUD 8.21 for each AUD 1 invested into running the program. This study shows the importance of considering companion animals as part of the family unit and supporting people experiencing a crisis to keep their companion animal.

**Abstract:**

Companion animals play a central role in many families and are especially valued by those who are socially isolated. Crisis situations such as acute hospitalizations, homelessness, and natural disasters can make it difficult to preserve the human–animal bond and can result in animals being surrendered or euthanized. Social support programs like the RSPCA NSW Emergency Boarding and Homelessness program support people experiencing crisis situations with emergency pet boarding, access to veterinary treatment and individualized case management. This study aimed to estimate the social return on investment (SROI) for this program using the standard SROI methodology. In-depth interviews were conducted with 13 program stakeholders and questionnaire responses were received from 29 program clients. Outcomes were quantified for four stakeholder groups: program clients, client’s animals, RSPCA Inspectors, animal pounds, and shelters. Clients and their animals experienced the bulk of the benefit from the program, estimated to have a combined value of over AUD 5 million for the 2020–2021 financial year. The estimated social return on investment was AUD 8.21 for each AUD 1 invested. The study demonstrates that keeping people together with their companion animals or ensuring they are reunited as soon as possible can reduce stressors and improve outcomes for people and animals.

## 1. Introduction

Companion animals play a central role in many families, providing their humans with companionship, affection, and someone to nurture and love. In Australia, more than two in three households have companion animals, mostly dogs and cats [1]. However, some circumstances make it difficult to preserve the human–animal bond and these circumstances often affect those for whom the human–animal bond is of the greatest value [2]. Crisis situations such as acute hospitalizations, homelessness, and natural disasters can necessitate families being separated, which can cause considerable stress and anxiety for both humans and animals and can lead to animals being relinquished to animal pounds or shelters or even being euthanized.

There are more than 116,000 people experiencing homelessness on any given night in Australia, a rate that is increasing [3]. Homelessness is associated with an increased risk of psychiatric illness, substance abuse, poorer physical health, and reduced life expectancy [4]. For those experiencing homelessness, animals provide companionship and familial connection, warmth, and comfort [5]. Having a companion animal is associated with motivation, resilience, responsibility, self-care, connection with others, and sense of purpose and decreases feelings of loneliness, risk of depression, and suicide [5]. In addition, despite misconceptions to the contrary, in our experience, the animals of those experiencing homelessness can enjoy excellent welfare, having constant companionship, freedom, and a stimulating environment. However, caring for an animal can prevent help-seeking behaviour, can limit access to public transport, employment, and medical care and can result in people being refused housing [6]. It is common for those experiencing homelessness to choose to continue to live outdoors rather than be housed without their animal companions [7]. Hence, having a companion animal can make it more difficult to seek safe accommodation. In addition, more than 275,000 people were admitted to hospital overnight for mental health in 2019–2020 in Australia, the average length of stay in NSW being 18 days [8], leaving many companion animals with nowhere to go.

Those experiencing a crisis such as homelessness or mental illness are more likely to be socially isolated and lack a support network [9]. As such, caring for companion animals can be an important barrier to people seeking help for themselves [10]. Animals left behind can cause people experiencing crises more worry, anxiety, and depression, while concern for their animal’s safety can negatively impact a person’s recovery [2,11].

The RSPCA NSW Emergency Boarding and Homelessness program supports people experiencing homelessness and various other crises including acute hospitalizations for mental or physical health conditions or natural disasters with their companion animals. The animals of those experiencing crises can be left in dangerous situations or without care while their owners access refuge or treatment. While separation from their companion animals might be unavoidable for a period, being able to reunite the family unit when possible is invaluable [2]. Companion animals provide support, comfort, and encouragement at times when these are of most value [12].

Social Return on Investment (SROI) is an evaluation approach based on methodologies used in economics, accounting, and social research [13]. It assesses how change is created by measuring social outcomes as experienced by key stakeholders. The participatory approach allows stakeholders including program participants and partner organisations to directly contribute to documenting changes due to a program and how outcomes are valued. The SROI approach is built on seven principles: (1) involve stakeholders; (2) understand what changes; (3) value the things that matter; (4) only include what is material; (5) do not overclaim; (6) be transparent; (7) verify the results. Social value is calculated by placing a financial value on the changes that occur for stakeholders using financial proxies. SROI also considers what would have happened anyway and change that is attributable to other actors.

Evaluating social support programs is crucial to ensure their effectiveness and impact. It allows policymakers and program administrators to assess whether the intended outcomes are being achieved and whether resources are being used efficiently. Evaluation helps identify strengths, weaknesses, and areas for improvement, leading to evidence-based decision-making and program refinement. As such, this study aimed to determine the social return on investment of the RSPCA NSW Emergency Boarding and Homelessness program, using the SROI methodology established by Social Value International [13].

## 2. Materials and Methods

### 2.1. The RSPCA NSW Emergency Boarding & Homelessness Program

The RSPCA NSW Homelessness & Emergency Boarding program supports pet owners experiencing a range of difficulties, enabling them to keep their animal companions while they focus on their own safety, recovery, or treatment. The program helps to keep animals together with their family in the long term by providing case management, pet boarding, assistance with transportation, assistance to access veterinary treatment, and other support as needed.

Individuals are eligible for the program if they are in hospital or temporary accommodation that does not allow pets and they have no support network (friends or family) who can help. Clients either self-refer or are referred by the RSPCA Inspectorate, police, hospitals, housing, or other social support services.

RSPCA NSW program case workers collaborate with other housing and social support services to ensure clients can access supports they need to find suitable accommodation so they can be reunited with their animals as soon as possible.

### 2.2. Identifying Stakeholders

Ethical approval for this study was granted by the University of Sydney Human Research Ethics Committee (protocol number: 2020/856).

The stakeholders for the RSPCA NSW Emergency Boarding and Homelessness program were identified through interviews with RSPCA NSW Community Programs staff who are involved in service design, planning, and delivery. Only stakeholder groups deemed material were included in the SROI calculation. The materiality of stakeholders was determined through discussion with RSPCA NSW Community Programs staff and validated during stakeholder interviews.

### 2.3. Stakeholder Engagement

A theory of change describes how certain resources (inputs) are used to deliver activities (outputs) that result in outcomes and is central to SROI analysis. A workshop was held with key staff from RSPCA Programs to review program objectives, identify and value inputs, define the services offered (activities), identify stakeholders to engage, and draft a theory of change for the RSPCA NSW Emergency Boarding and Homelessness program.

#### 2.3.1. Interviews

A series of in-depth stakeholder interviews were conducted with current and previous clients of the program, RSPCA Inspectors, and representatives from partner organisations, e.g., NSW Police, domestic violence services, and local council staff. Interviews were conducted to understand what changes because of the three RSPCA NSW Community Programs (including positive and negative, intended, and unintended), the nature of the change, and who experience the change. These interviews, along with the theory of change workshop were also used to establish the materiality of various stakeholder groups and explore ideas of attribution (how much of change is due to other influences), deadweight (what would have happened anyway), benefit period (how long does the change last), and drop-off (how much does the value diminish over time). Interviews also invited participants to reflect on other stakeholders that might have experienced change because of the program. Client interviews were used as proxy to identify and reflect on changes experienced by their animals.

Current and previous RSPCA NSW Community Programs clients were invited to participate in interviews by their RSPCA case workers. Participants were chosen based on their diversity of experiences. Interviews were semi-structured and followed a planned interview guide for clients, RSPCA NSW staff, and external stakeholders, respectively (Appendix A). They were conducted by one of the authors (GM) by telephone, video call, or face-to-face based on the preferences of the individual.

Interviews were audio-recorded and transcribed using the transcription tool in Microsoft Word. The interview audio was reviewed manually by GM to correct errors in the transcription. Thematic analysis of interview transcripts was conducted using an inductive approach. Transcripts were coded in NVivo independently by GM and JR. Emergent themes were discussed jointly by the research team to ensure the validity and reliability of the findings. Transcripts were re-reviewed frequently during the analysis to ensure the interpretation of themes remained true to the interview transcripts.

#### 2.3.2. Questionnaires

A questionnaire was used to determine outcome incidence and quantify the amount of each outcome experienced by clients of RSPCA NSW Emergency Boarding and Homelessness program (Appendix B). The questionnaire was also used to determine the value of the counterfactual (attribution, deadweight, and displacement) and determine the benefit period and drop-off for each outcome. Previous and current RSPCA Community Programs clients were invited by their caseworker to complete the questionnaire, which was hosted on REDcap© (Vanderbilt University, Nashville, TN, USA). Clients were also asked about outcomes experienced by their animals.

Clients were asked to choose the extent to which they agreed with a set of statements about the nature of the change they experienced because of their participation in the program on a 5-point Likert scale from A lot WORSE to A lot BETTER. Clients were also asked open-ended questions about what they considered to be the most important change they experienced, and what would have been different if they were not able to access the RSPCA NSW program.

Clients that responded a little BETTER or a lot BETTER on the corresponding indicator question were considered to have experienced an outcome. Where there was more than one indicator question, a client was considered to have experienced that outcome if they responded a little BETTER or a lot BETTER to any of the indicator questions. Client responses to the open-ended questions were coded and responses that showed the client had experienced an outcome were also included.

### 2.4. Analysis and Modelling

The value of outcomes was explored through the in-depth stakeholder interviews using a relative value and stated preference approach. Desktop research was undertaken to identify suitable financial proxies for each outcome experienced by each stakeholder and to determine appropriate discount factors (deadweight, displacement, benefit period, and drop off).

The value of program inputs for the 2020–2021 financial year was determined by examining program financial records. Interviews with program staff identified additional donated and in-kind inputs.

The social value of the RSPCA NSW Emergency Boarding program was calculated by assigning financial proxies to represent the social value created by each outcome as experienced by that stakeholder group. Financial proxies were selected based on client interviews and desktop research. The social value associated with each outcome was calculated by multiplying the outcome incidence by the value of the financial proxy. The relative value of outcomes was determined through client interviews and questionnaire responses (both Likert scale and open-ended questions). Clients were asked during interviews to state the value of changes experienced. Clients were also asked in the questionnaire “What are the most valuable changes that have happened for you as a result of your experience with RSPCA?”.

Discount factors for each outcome including deadweight (the amount of the outcome that would have happened anyway), attribution (how much of the outcome was caused by other organisations or people), displacement (how much the outcome displaced other outcomes), and drop-off (how much the value of an outcome decreases over time) were estimated based on client interviews and questionnaire responses to the questions “have you received support from other agencies/people?”, “how much of this difference in your life is due to RSPCA? Can you estimate a %?”, and “what do you think would be different for you now if you had not accessed assistance for your animal/s through RSPCA?”.

The results of the SROI analysis were presented to RSPCA Programs staff, interview participants, and current clients for validation and feedback before finalizing the model. A conservative approach was adopted for decisions on data and assumptions used in the SROI calculation. Values in this report might understate the actual value created.

## 3. Results

Note: the names of all research participants and their animals have been changed to protect their identities.

The RSPCA NSW Emergency Boarding and Homeless program assisted 627 animals belonging to 259 clients in the 2020–2021 financial year. Of these, 446 animals accessed emergency boarding or foster care, spending a total of 12,206 days in care. Veterinary treatment was also facilitated for 296 of these animals, worth AUD 71,291 in total.

In depth interviews were conducted with 3 program clients, 8 RSPCA inspectors, and 2 external stakeholders, one from NSW Police and one from a local council. Questionnaire responses were received from 29 program clients. Interviews and questionnaire responses were used to refine the RSPCA NSW Emergency Boarding and Homelessness program theory of change (Figure 1).

The SROI investigated outcomes for four stakeholder groups who were determined to be material: (1) program clients; (2) animals of program clients; (3) RSPCA Inspectors; (4) animal pounds and shelters (Table 1).

### 3.1. Evidencing Outcomes

#### 3.1.1. Outcomes Experienced by Clients

The importance of the human–animal bond to clients of this program cannot be overstated; the support, affection, and encouragement provided by clients’ animal companions helped them to feel safe, motivated, needed and in many cases, gave them a reason to live (Figure 2).

Clients who were assisted by the RSPCA NSW Emergency Boarding and Homeless program were able to extend or enhance their bond with their companion animal and because of the support received clients experienced improved mental health and wellbeing. These were the outcomes most valued by clients and the ones experienced most frequently. The language used by clients both in interviews and in open-ended questionnaire responses indicated that in some circumstances their animals were their reason to live or the thing preventing them from dying by suicide.


*“That I got to be able to still have my cat because there are times in my life where my cat has saved me [from suicide].”*
Stacy—program client.


*“[My cats] are the reason I feel carpet under my feet in the morning. Over the years my life got to the point where I’ve lost so much, and not just material stuff and money, but all the other shit that goes with it, and they were my anchor. They made me come home every night. Be there to feed them by 6 o’clock every. Single. Day. I’m up every single morning to put food in their bowls. Their bowls are washed out religiously. These are the things I’ve got to do every day. That’s what the cats mean to me. They basically took the place of an antidepressant.*

*When you’re in the situation I was in—a crisis situation—you’re not connected to anything. You’re quite alone and even though there were services around me the cats kept me focused on what I needed to do. Do you know what I mean? It wasn’t all lost. The girls were coming with me and that made it sort of better. [When I knew the cats were safe] it was just like “I’m good now. I can do anything.” It was just really, really important that I had them there and I knew that they were coming back. Yep 100%, that was just everything.”*
Amber—program client.


*“He means everything to me. He is my life. He is the only one that I have in my life. He keeps me sane. I don’t know what I would do or where I would be without him. He is very special to me. He’s not just a pet, he is my companion and my support. He is my everything. I cannot see myself living without him. I am one of the lucky ones to have a dog.”*
Kimberly—program client.


*“[Without RSPCA] it would be a lot different. My life would be a little bit empty without them because I’ve spent the last 10 years looking after them. My life would feel empty without them. Like if you have animals and cats in your life and then all of a sudden you don’t have them, because you had an accident and went to hospital, and the cats get re-homed, you know that would be a devastating thing to go through. I would miss them terribly. And even small jobs like cleaning the kitty litter and stuff like that. There’s nothing that can replace the happiness having a cat can bring you. I’d hate it and the animals wouldn’t like it either.”*
Scott—program client.

Clients also experienced reduced financial stress and social isolation, improved safety, improved physical health, and were better able to access care for themselves (such as drug and alcohol programs, mental health care, physical rehabilitation, or respite).


*“[The most valuable thing was] knowing there is help out there, as I am a single mum and do not have a job, so I could not afford the vet treatment. I would have been forced to surrender her, but because of RSPCA’s help we could stay together as a family and I am so grateful to be able to pay this off, it means everything to me. RSPCA will never know how much I appreciate what they have done for us.”*
Diana—program client.

All except one of 29 clients who completed the questionnaire (97%) reported experiencing at least two positive outcomes. The median number of positive outcomes experienced was five (average 4.8). The outcomes experienced most frequently by clients were an extended or enhanced human–animal bond (27 of 29 clients, 93%), improved mental health and wellbeing (26 of 29 clients, 90%), decreased financial stress (22 of 29 clients, 76%), and increased social inclusion and decreased social isolation (21 of 29 clients, 72%; Table 1).

#### 3.1.2. Outcomes Experienced by Clients’ Animals

Clients who completed the questionnaire were assisted with up to two animals; 21 clients had one dog, two had two dogs; six clients had one cat, and two had two cats; two clients were assisted with one dog and one cat. In total 35 animals were assisted by the program for clients who responded to the questionnaire.

Animals belonging to clients of the RSPCA NSW Emergency Boarding and Homelessness program faced losing their human family, often also losing their lives if they were not able to access support.


*“For the first time I had some actual choices. When you are in crisis, your animal’s welfare is so at risk.”*
Jennifer—program client.


*“[Without RSPCA] it would have been the end of my dog as I would have had to have her euthanized.”*
Brian—program client.


*“[Without RSPCA] my kids would have been heartbroken as we would have had to surrender her.”*
Diana—program client.

All 29 clients who completed the questionnaire reported positive outcomes experienced by their animals. The outcomes experienced by the client’s animals that were included in the model were (1) improved wellbeing as a result of preserving or improving the human–animal bond (experienced by 32 of 35 animals, 91%); (2) access to safe accommodation (experienced by 25 of 35 animals, 71%); (3) improved physical health (experienced by 25 of 35 animals, 71%; Table 2).

#### 3.1.3. Outcomes Experienced by RSPCA NSW Inspectors

Animals that are assisted by the RSPCA NSW Emergency Boarding and Homelessness program might otherwise be referred to the RSPCA Inspectorate for several reasons. Animals might be abandoned at a property without care when a client is taken to hospital or becomes homeless, animals might also be referred for investigation by the Inspectorate for possible neglect or failure to provide veterinary care. Through the Inspectorate, these cases are investigated and potentially prosecuted under the *Prevention of Cruelty to Animals Act NSW* (1978), the animals would be seized and taken into RSPCA custody, separated from their families, and housed in an animal shelter. Without community assistance programs like the RSPCA NSW Emergency Boarding and Homelessness program, taking animals into custody can be the only option available to Inspectors to ensure the safety of the animals involved. Hence, the RSPCA Inspectorate was determined to be an important stakeholder.

Two outcomes experienced by RSPCA Inspectors were considered material and included in the model (Table 3). While both these outcomes were experienced by all the Inspectors interviewed, a conservative outcome incidence of 80% has been used in the model in recognition of the small sample size.


*“It’s a huge relief to have programs available to assist. If I had more cases that programs couldn’t assist with, I’d feel stressed, and it would be quite difficult. I can go home and sleep at night and not worry that I left the dog in an environment that wasn’t ideal and there’s no monitoring. Or I’m going to have to take someone through a court system for something that I know ahead of time, it’s going to go under the Mental Health Act. You have to question why you’re taking that route in the first place. But if there are significant animal welfare issues and you can’t leave the animal, you’ve got no other choice. I mean, I would hate to be plagued with that on my mind because my mental health will start to be affected and I’d probably end up needing assistance myself.”*
RSPCA Inspector.

#### 3.1.4. Outcomes Experienced by Animal Pounds and Shelters

The RSPCA NSW Emergency Boarding and Homelessness program prevented animals arriving at animal shelters or pounds by reducing the number of animals abandoned without care or surrendered by their owners. Abandoned and surrendered animals often experience considerable distress when separated from their families and placed in a shelter environment. They are not always medically or behaviourally suitable to be rehomed, in which case they might be euthanized. Those who are suitable to be rehomed can require substantial investments of time and resources to ensure they are behaviourally and medically ready to be adopted.

Animal pounds and shelters were determined to experience two material outcomes that have been included in the model based on interviews with RSPCA NSW Community Programs staff, RSPCA NSW Inspectors, and program clients. The outcome incidence was determined based on client responses to the question ‘What would have happened to your animal/s if RSPCA could not assist you?’ The two outcomes included were (1) fewer animals abandoned without care; (2) fewer animals surrendered for rehoming (Table 4).


*“They would’ve taken my dog away from me if I couldn’t find help.”*
Nathan—program client.


*“I was very stressed at the time and financially strained. [Without RSPCA] I would have had to surrender my animals.”*
Angela—program client.

#### 3.1.5. Unintended and Negative Outcomes

No clients reported negative outcomes as a result of the RSPCA NSW Emergency Boarding program through in-depth interviews. In the client questionnaire, six of 29 clients (21%) selected ‘a little worse’ or ‘a lot worse’ for one or more of the Likert-scale outcome questions (Figure 3). In addition, one client described a negative experience through the open-ended question ‘What are the most valuable changes that have happened for you as a result of your experience with the RSPCA?’.


*“I was not happy at all with the services from RSPCA… they didn’t ask my permission before treating my dog.”*
Veronica—program client (Veronica’s dog was collected by an RSPCA Inspector after being left unattended at a property while Veronica was in hospital).

Negative outcomes expressed through the Likert-scale questions were included in the SROI model.

### 3.2. Valuing Outcomes

Financial proxies selected for each outcome and the rationale for their selection are detailed for each stakeholder group in tables in Table 5, Table 6, Table 7 and Table 8. Clients struggled to place a financial value on changes attributed to their participation in the RSPCA NSW Emergency Boarding and Homeless program, especially when discussing the value of maintaining the human–animal bond. There was no traded good that adequately reflected what their companion animal meant to them.

When asked “what are the most valuable changes that have happened for you as a result of your experience with RSPCA?” the outcomes most often mentioned by clients were (1) improved mental health and wellbeing (reported by 10 of 29, 34% of clients); (2) extended or enhanced human–animal bond or improved animal health or wellbeing (reported by 8 of 29, 28% of clients); (3) decreased financial stress (reported by 6 of 29, 21% of clients); (4) increased social inclusion/decreased isolation (reported by 5 of 29, 17% of clients).

### 3.3. The Counterfactual

#### 3.3.1. Deadweight

Deadweight is a measure of the amount of outcome that would have happened even if the activity had not taken place [13]. Clients were asked to reflect on what would have happened or how their life would be different if they had not accessed the program in both in-depth interviews and the client questionnaire. This was also explored through in-depth interviews with RSPCA Inspectors and other external stakeholders.

Almost all clients reported that they would have experienced some negative outcome if they had not been able to access the program (26 of 29, 90%) based on their response to the open-ended question ‘What do you think would be different for you now if you had not accessed assistance for your animal/s through RSPCA?’ in the client questionnaire. Two clients did not answer this question.

The most frequently reported negative outcomes were: (1) losing their animal/s (whether surrendered to a pound or shelter, rehomed, abandoned, or euthanized) (reported by 16 of 29 clients, 55%); (2) increased stress or worse wellbeing (reported by 9 of 29 clients, 31%); (3) worse animal health or wellbeing (reported by 7 of 29 clients, 24%); (4) worse financial stress (reported by 7 of 29 clients, 24%).

A total of 2 of 29 clients (7%) mentioned other options that might have provided some of the positive outcomes experienced as a result of their participation in the RSPCA program. The two alternatives suggested were seeking emergency boarding for their animal with friends or boarding their animal through a private boarding facility:


*“I would have found a friend to look after him.”*
Jason—program client.


*“I would have been forced to find the money for private boarding services.”*
Angela—program client.

Hence, a deadweight of at least 7% has been applied for all outcomes. We consider this conservative given the demand for the program, the experiences described by RSPCA Inspectors, and the lack of alternatives available as described by clients and external stakeholders in interviews. A deadweight of up to 25% has been applied for some outcomes, e.g., ‘Improved wellbeing as a result of preserving the human–animal bond’ and ‘Fewer animal surrendered by their owners’ acknowledging that these outcomes might have occurred for some clients without receiving support with their companion animals (Table 9).

#### 3.3.2. Attribution

Attribution is an assessment of how much of the outcome was caused by other organisations or people [13]. For outcomes relating directly to service provision attribution of 100% was applied. Attribution for other outcomes varied according to the nature of the outcome. Some outcomes for example ‘Improved mental health and wellbeing’, ‘Improved physical health’, and ‘improved personal safety’ as experienced by clients were considered to have a relatively large potential contribution from other sources such as accessing specialist homelessness or mental health services and crisis accommodation, hence, a discount of up to 75% has been applied for these outcomes (Table 9).

We consider this discount to be conservative. Several clients mentioned not receiving assistance from any other agencies. For example, when asked about the support received from other agencies or organisations through their experience two interview clients responded:


*“No. No, I don’t know anybody.”*
Robert—program client.


*“I got online so you know like 12 months through the whole disaster, the drug raids and bloody jail and everything like that. And the cheapest thing I could find was over in [a distant suburb] and it was like $450 a week… for 5 cats per week was around 700 bucks.”*
Amber—program client.

#### 3.3.3. Displacement

Displacement is an assessment of how much the outcome has displaced other outcomes, for example, if the activities of the program prevent people experiencing the same changes somewhere else [13]. For most outcomes, the stakeholder experiencing the outcome does not preclude other stakeholders from also experiencing the outcome. Hence, a displacement of 0% has been applied for most outcomes for most stakeholder groups.

One outcome where some displacement has been included is ‘Extended or enhanced human–animal bond’. Supporting clients to keep their animal/s displaces potential adopters of these animals from experiencing this outcome if the animal were rehomed or surrendered to a pound or shelter. Likewise, animals rehomed might ultimately have similarly beneficial relationships with a new family, notwithstanding the stress and trauma this rehoming process would cause to the animal in the short to medium term. Hence, a displacement of 25% has been included for these outcomes (Table 9).

### 3.4. Benefit Period and Drop-Off

A benefit period is the amount of time an outcome is applied in the SROI model, while drop-off reflects the deterioration of an outcome over time [13]. The benefit period for outcomes ranged from one to nine years and varied depending on the nature of the change. Some outcomes applied only for the period the client accessed the program, hence, a benefit period of one year was used. For example, ‘decreased financial stress’ was experienced once at the time of participating in the program.

Other outcomes continue for the remainder of the life of the companion animal, for example, ‘extended or enhanced human–animal bond’ as experienced by clients and ‘improved wellbeing as a result of preserving or improving the human–animal bond’ as experienced by clients’ animals. Other benefits might endure for much longer, for example, the benefits to mental health and wellbeing of supporting a person in crisis; however, a benefit period of one year has been applied so as not to overclaim.

Likewise, the drop-off value varied between outcomes depending on the nature of the outcome. For some outcomes, there was no drop-off over time, for example, the value of preserving the human–animal bond as experienced by the animal. Rather than decreasing in value over time, this relationship strengthens and becomes more important to the animal, while the animal also becomes less able to find a different home with age. A 5% drop-off was applied for ‘extended or enhanced human animal bond’ as experienced by both clients and their children, however this is likely to be conservative (Table 9). In many instances the bond a person feels for their companion animal continues to strengthen and become more meaningful throughout the animal’s life, indeed relationships with companion animals can be as important as those with a ‘significant other’ or ‘soul mate’ [24].

NB to be conservative when claiming, the net present value of outcomes with a benefit period of nine years has only included the value of these outcomes for the first three years.

### 3.5. Program Inputs

Inputs for the RSPCA NSW Emergency Boarding and Homelessness program, which supported 259 people with 627 animals in the 2020–2021 financial year, were determined through interviews with RSPCA NSW Community Programs staff and examination of program financial records. Values are based on actual costs for the 2020–2021 financial year. Where products and services have been donated or discounted the full market value has been included in the analysis. The total inputs were valued at AUD 642,489 including people costs, veterinary treatment, animal boarding, volunteer time, and other non-people and overhead costs.

### 3.6. The Social Return on Investment

The benefits experienced by clients and their animals accounted for 95% of the total value created for all stakeholders, which was assessed to be worth over AUD 5,000,000 (Table 10).

The greatest share of benefits was created for program clients, valued at AUD 2,887,569 for the 2020–2021 financial year or an average social value generated per client of AUD 11,149. Program clients’ animals were the stakeholder group experiencing the next largest share of social benefit, with a total of AUD 2,151,633 for all animals, or AUD 432 per animal assisted. Overall, the RSPCA NSW Emergency Boarding and Homelessness program was estimated to have generated AUD 8.21 in social value for each AUD 1 invested into running the program (Table 11).

## 4. Discussion

This SROI demonstrates the RSPCA NSW Emergency Boarding and Homelessness program provides a considerable social return on investment, with most of the benefit shared between clients and their companion animals. The bulk of the benefit can be attributed to preserving the human–animal bond and to improvements to the mental health and wellbeing of both human and animal family members. This study demonstrates the value of using SROI methodology to provide a comprehensive and quantifiable analysis of both financial and non-financial outcomes for a social program, enabling a more holistic understanding of its impacts.

The RSPCA NSW Homelessness and Emergency Boarding program supports people experiencing the greatest challenges of their lives, a time when they have their greatest need for social support. This study, consistent with previous studies, demonstrates that companion animals can be critical for crisis recovery, providing a sense of safety, family, purpose, routine, and a source of unconditional, judgement free love, affection, and understanding [2,5]. Companion animals are often present with their person as they experience some of their hardest days. Companion animals can also facilitate connections with others, helping improve a person’s integration into a community and social network [25,26]. This support is even more valuable to those who are socially isolated [27]. Most importantly, as noted repeatedly by clients in this study and as previously reported, companion animals provide a sense of grounding and give life post-crisis purpose and direction—a reason to get up in the morning or keep going [2,10,28,29].

For those experiencing homelessness, physical or mental health crises, or any other crisis, the support and encouragement they receive from their animal companions can be literally lifesaving. Clients of the RSPCA NSW Homelessness and Emergency Boarding program often experience almost complete social isolation, with their companion animals providing their only source of social support [2]. Social isolation and loneliness are serious risk factors for suicide [30]. Conversely, access to social support is associated with decreased risk of suicide [31]. Interpersonal and affective relationships can give meaning to life and mitigate against suicide, and these can include relationships with animals [28]. Psychological elements of social support such as empathy, love, compassion, belongingness, usefulness, and acceptance can be provided by animal companions where individuals lack a human social network [2]. Hence, supporting socially isolated individuals to preserve their relationships with their companion animals should be prioritized, especially for those with known mental health challenges and increased suicide risk—their animal can be the only thing keeping them going.

Many people assisted by the RSPCA NSW Homelessness and Emergency Boarding program are referred to the program by the RSPCA NSW Inspectorate, often after receiving a complaint of animal cruelty or neglect from a concerned neighbour or member of the public. These complaints to the Inspectorate can be the first indication a socially isolated person with an animal is not coping. Opportunities for enforcement agencies, especially police, to act as referral pathways for social support have been noted in the literature [32,33], and this could help enforcement personnel avoid experiencing moral distress when they encounter situations that would be more effectively managed with social support [34]. Building rapport and trust with an isolated person by supporting them with their animal can lead to them accepting other services such as housing, social workers, or mental health support that might otherwise have been refused [35]. Our findings demonstrate that in addition to benefiting the person in question, making appropriate referral pathways available is also important for the mental health and wellbeing of Inspectors and allows for enforcement agencies to operate more efficiently. This highlights the importance of interagency collaboration, especially between animal welfare enforcement agencies and social support agencies, and the unique role of animal-specific social support programs to connect people who are isolated with the services they need.

Caring for companion animals can be an important barrier to seeking help, especially for those with a limited support network. Accessing health care, especially in-patient treatments, and accessing temporary accommodation can be difficult or impossible for those without someone to care for their animals while they are away. The possibility of having to relinquish their animals, or worrying about their care while staying with neighbours or family can be enough of a deterrent to cause people to delay or avoid seeking the care they need. This has been noted as important amongst older people, those with disabilities, those experiencing domestic and family violence, and those experiencing homelessness [2,9,29,36]. Our study demonstrates that providing support for people’s companion animals removes some of these barriers, resulting in improved access to support for themselves, and better physical and mental health outcomes.

There is currently a major shortage of crisis accommodation options for animals and demand for these services far outstrips their capacity. This is compounded by a general shortage of animal-friendly housing including private rentals, temporary accommodation, and crisis refuges [37,38,39]. For those for whom private pet boarding is cost-prohibitive, surrender, rehoming, or euthanasia of their animal companions can be their only option. However, cost is not the only barrier [40]. Those in greatest need of support with their animals often experience multiple barriers to accessing private pet boarding such as a lack of transport, incapacity (for those who are hospitalized for being physically or mentally unwell), even their animal having medical problems, or not being up to date with vaccinations. Wholistic programs like the RSPCA NSW Emergency Boarding and Homelessness program need to be expanded to keep animals together with their person. As we have shown, animal shelters and pounds directly benefit when people are assisted to retain their companion animals. Hence, these organisations are well positioned to provide animal-specific social support services and should do so where possible.

While the study provides valuable insights into the social value created by the RSPCA NSW Emergency Boarding and Homelessness program, it also has some limitations. The relatively small sample size might not have captured the full range of experiences of clients and their perspectives related to the program. In addition, there is inherent subjectivity in assigning monetary value to social outcomes. In this study, clients were not asked how RSPCA NSW could have provided a better service—this is a potential limitation. Future qualitative research could explore how various program activities contribute to outcomes experienced by stakeholders and how these can be optimized for clients with different needs.

## 5. Conclusions

Programs that support those experiencing homelessness, physical, or mental health crises or any other crisis situation with their companion animals are a vital component of aiding recovery. When their animals are safe and cared for, people can focus on their own recovery and safety. For those who have been through challenging times with their animals, having survived together creates an even stronger bond between humans and animals. Hence, being able to live and recover together is essential. Keeping people together with their companion animals or ensuring they are reunited as soon as possible can reduce stressors and improve outcomes for people and animals.

There is a need to expand wholistic animal-specific social support programs like the RSPCA NSW Emergency Boarding and Homelessness program. There is also a need for greater integration of services that consider animals as part of a person’s family unit and part of their social support system. A companion animal can be a person’s greatest strength and should be considered as such by enforcement agencies, medical providers, mental health services, and social workers.

## Figures and Tables

**Figure 1 animals-13-02264-f001:**
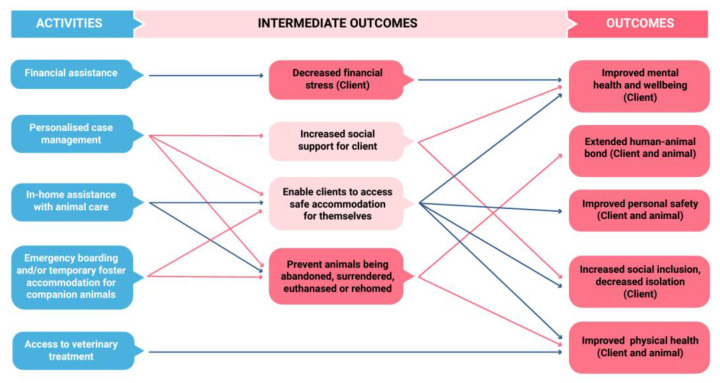
How change is created as a result of the RSPCA NSW Emergency Boarding and Homelessness program.

**Figure 2 animals-13-02264-f002:**
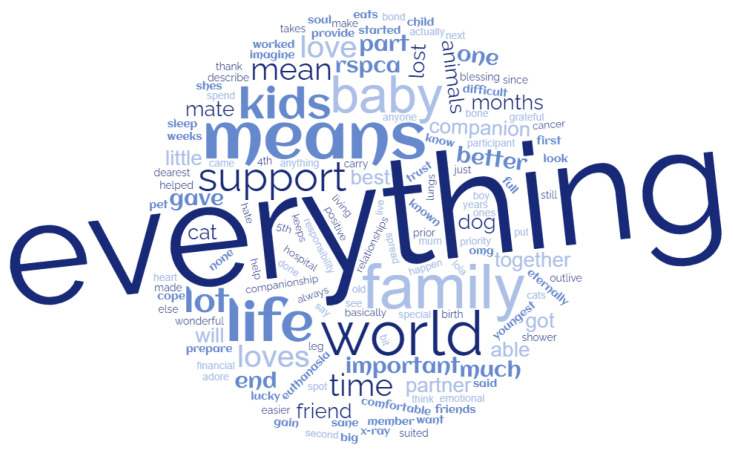
Word cloud of program client responses to the question “what does your animal mean to you?” (WordClouds.com, accessed 9 November 2022).

**Figure 3 animals-13-02264-f003:**
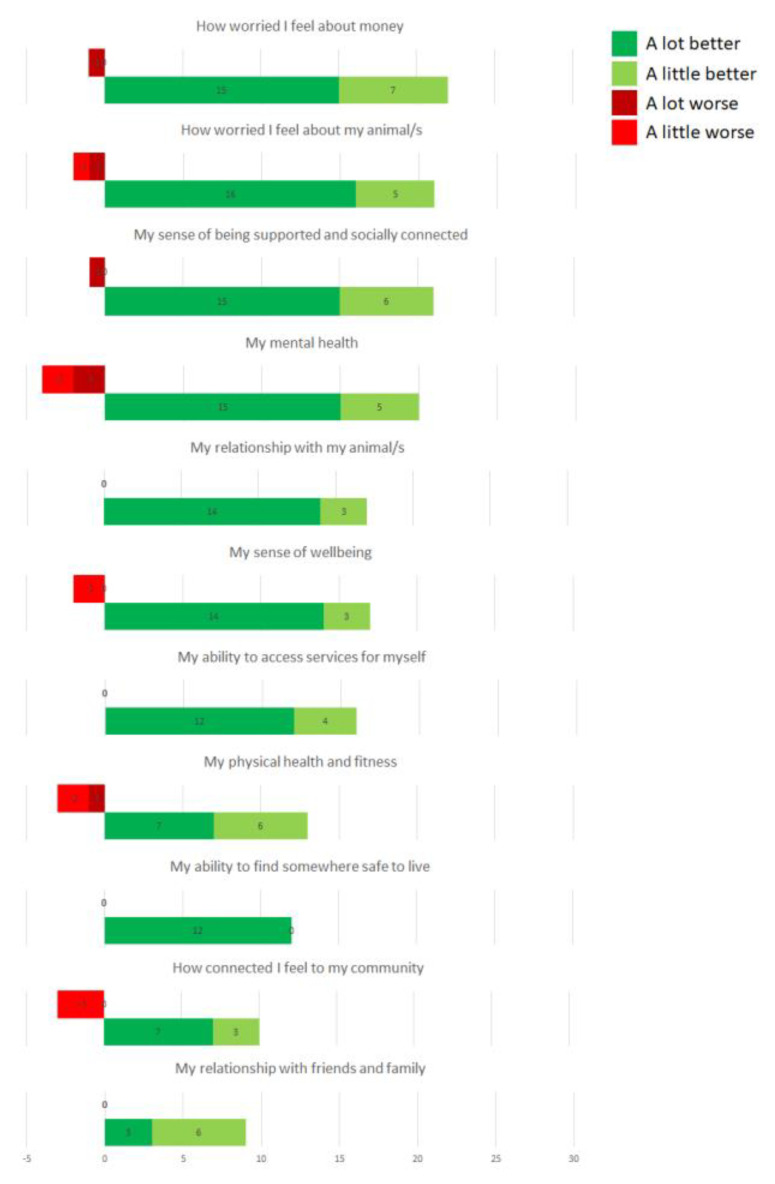
RSPCA NSW Emergency Boarding and Homelessness program client’s responses to Likert scale questions in client questionnaire.

**Table 1 animals-13-02264-t001:** Rationale for inclusion and exclusion of stakeholders in social return on investment of the RSPCA NSW Emergency Boarding and Homelessness program based on client questionnaire responses.

Stakeholder	Included	Reason for Inclusion/Exclusion
Program clients	Yes	Group expected to directly benefit the most from the activity
Companion animals of clients	Yes	Benefit directly from program activities
RSPCA NSW Inspectorate	Yes	Clients case-managed through the RSPCA NSW Domestic Violence program are often diverted from being managed through the RSPCA Inspectorate
Animal pounds and shelters	Yes	Animals accessing services through the Emergency Boarding program are less likely to be surrendered to animal pounds and shelters
Broader community	No	Changes not material
Family and friends of clients	No	Changes not material
Other service providers	No	Changes not material
NSW Health	No	Improved physical health and fitness experienced by program clients might have resulted in less health expenditure. Changes experienced by NSW Health are likely to be material but were beyond the scope of this project to quantify.
Court system	No	Clients referred to the RSPCA NSW Emergency Boarding and Homelessness program via the RSPCA Inspectorate might otherwise be subject to prosecutions for animal cruelty or neglect under the *Prevention of Cruelty to Animals Act 1979* (NSW). Changes experienced by the NSW Court system are likely to be material but were beyond the scope of this project to quantify.

**Table 2 animals-13-02264-t002:** Outcome incidence experienced by clients of the RSPCA NSW Emergency Boarding & Homelessness program based on client questionnaire responses.

Outcome	Incidence
Extended or enhanced human–animal bond	27 of 29 (93%)
Improved mental health and wellbeing	26 of 29 (90%)
Increased social inclusion/decreased isolation	21 of 29 (72%)
Decreased financial stress	22 of 29 (76%)
Improved access to care for themselves	17 of 29 (59%)
Improved personal safety	12 of 29 (41%)
Improved physical health	11 of 29 (38%)

**Table 3 animals-13-02264-t003:** Outcome incidence experienced by the animals of clients of the RSPCA NSW Emergency Boarding and Homelessness program based on client questionnaire responses.

Outcome	Incidence
Improved wellbeing as a result of preserving or improving the human–animal bond	32 of 35 (91%)
Improved physical health	25 of 35 (71%)
Access to safe accommodation	25 of 35 (71%)

**Table 4 animals-13-02264-t004:** Outcome incidence experienced by RSPCA NSW Inspectors because of the RSPCA NSW Emergency Boarding and Homelessness program.

Outcome	Incidence
More time available to pursue genuine animal cruelty offenses	80%
Improved mental health	80%

**Table 5 animals-13-02264-t005:** Outcome incidence experienced by animal pounds and shelters because of the RSPCA NSW Emergency Boarding and Homelessness program.

Outcome	Incidence
Fewer animals abandoned	9%
Fewer animals surrendered	37%

**Table 6 animals-13-02264-t006:** Valuing outcomes experienced by clients of the RSPCA NSW Emergency Boarding and Homelessness program.

Outcome	Proxy Description	Rationale	Value	Source
Improved mental health and wellbeing	Contingent valuation. The value of a statistical life year (AUD 222,000) adjusted for the loss attributable to generalised anxiety disorder—mild to moderate (disability weighting 0.17). Benefit applied for the average length of time an Emergency Boarding client’s animal/s were in care (51 days).	Clients’ companion animals provide an important source of comfort and companionship that is a consistent presence in their day-to-day life. Participation in the program enables clients to preserve this relationship, which profoundly improves wellbeing and was considered the most valuable change by clients. Hence, we equate this outcome with relieving mild anxiety.	AUD 5320	Value of a statistical life year: Australian Government, Department of Prime Minister and Cabinet [14] Disability weight: Australian Government, Australian Safety, and Compensation Council [15]
Extended or enhanced human–animal bond	Contingent valuation. The value of a dog life year for a companion dog with a receptive owner applied for the difference between the average age of an Emergency Boarding program dog (5 years) and a dog’s average life expectancy (13 years) [16]. Based on a value of USD 2400 per year of the dog’s life in 2019, converted to present value Australian dollars.	Clients described the depth of the bond they have with their companion animals; a bond that in many cases had been strengthened by their shared experiences of trauma. Their relationship with their companion animal was often their most valuable, even their only relationship. They also described the impact that losing this bond would have on their wellbeing.	AUD 3453	Carlson et al., 2019 [17]
Decreased financial stress	Observed spending on related goods. The difference in cost between the emergency boarding rate charged to RSPCA NSW Community Programs clients (AUD 10/day) and the cost of boarding through a private pet boarding facility (AUD 50/day) for the average duration an Emergency Boarding client’s animal/s were in care (51 days).	Clients experiencing this outcome avoid incurring upfront costs associated with providing safe temporary accommodation for their animals through private boarding facilities. Clients also receive discounted pet boarding through the program. In interviews, clients reported the alternative to accessing the RSPCA program would be to pay for pet boarding through a private boarding facility or veterinary hospital and that this would typically cost around AUD 50/day.	AUD 2058	Client interviews
Increased social inclusion/decreased isolation	Time use method. The average amount spent on recreation for six months. Based on the average weekly income for a one-parent family of AUD 1187 and the proportion of weekly income spent on recreation for families in the lowest income bracket (9%).	According to clients, participation in the Emergency Boarding program increased their social inclusion and decreased their social isolation including increasing their sense of being supported and connected to their community and improved relationships with family and friends. Hence, we use the amount spent on recreation as a proxy to represent improved social interactions and social connectedness. Six months was selected as the length of time this outcome was anticipated to last.	AUD 2778	Australian Bureau of Statistics [18]
Improved access to care for themselves	Observed spending on related goods. The cost of an allied health assistant (AUD 56.16/h) for one hour once per week for the average duration an Emergency Boarding client’s animal/s were in care (51 days).	Clients reported in interviews and questionnaire responses that the assistance provided by RSPCA enabled them to better access care and services for themselves, for example attending hospital, accessing mental health, and rehabilitation services. We consider a weekly session with an allied health assistant would provide similar benefits. Allied health assistants facilitate functional improvement and provide supports aimed at adjustments, adaptation, and building capacity for clients.	AUD 413	National Disability Insurance Agency [19]
Improved physical health	Observed spending on related goods. The typical cost of an annual gym membership.	Some clients reported experiencing improved physical health as a result of their participation in the RSPCA Emergency Boarding program. We determined that an annual gym membership would provide similar benefits to clients experiencing this outcome.	AUD 1140	Canstar Blue [20]
Improved personal safety	Observed spending on related goods. The cost of secure accommodation based on the median weekly rent for NSW 2020–2021 of AUD 466.25 per week for the average length of time Emergency Boarding client’s animal/s were in boarding or foster care (51 days).	Clients were better able to seek safety for themselves as a result of having somewhere safe to place their animals. In the context of people experiencing homelessness or medical or mental health crises, finding secure accommodation would be expected to provide a similar outcome for clients. Hence, the median weekly rent in NSW for the duration client’s animal/s were in care has been used as a proxy.	AUD 3427	NSW Government [21]

**Table 7 animals-13-02264-t007:** Valuing outcomes experienced by the animals of RSPCA NSW Emergency Boarding and Homelessness program clients.

Outcome	Proxy Description	Rationale	Value	Source
Improved wellbeing as a result of preserving or improving the human–animal bond	Observed spending on related goods. The cost of insurance premiums for a typical Emergency Boarding program animal (5-year-old Australian Cattle Dog Cross).	We have chosen the cost of insurance to reflect the value placed on ensuring an animal’s continued wellbeing by their owner. We consider this to be an outcome that continues for the life of the animal. In addition, this is a relationship that strengthens over time, increasing rather than decreasing as animals age.	AUD 1380	Choosi: Pet Insurance. Pet insurance comparison website [22]
Access to safe accommodation	Observed spending on related goods. The cost of private pet boarding for the average number of days an Emergency Boarding program animal remained in care (39 days).	RSPCA Emergency Boarding program clients’ animals access safe accommodation either in secure boarding facilities or with foster families. Accessing private pet boarding would provide a similar outcome for these animals hence the cost of private boarding has been used as a proxy. Clients reported in interviews that this would cost around AUD 50 per day through local boarding kennels or veterinary practices.	AUD 1925	Client interviews
Improved physical health	Observed spending on related goods. The average cost of veterinary treatment per animal that received veterinary treatment while under the care of the Emergency Boarding program.	Most animals participating in the RSPCA Emergency Boarding program accessed veterinary treatment, whether routine medical care or treatment or injuries or illness and as a result experienced improved physical health. Hence, the average cost of the veterinary treatment provided per animal receiving veterinary treatment through the program was used as the proxy for this outcome.	AUD 241	RSPCA NSW Community Programs records

**Table 8 animals-13-02264-t008:** Valuing outcomes experienced by RSPCA NSW Inspectors because of the RSPCA NSW Emergency Boarding and Homelessness program.

Outcome	Financial Proxy	Rationale	Value	Source
More time available to pursue genuine animal cruelty offences	Time use method. Value of an Inspectors time spent pursuing Emergency Boarding programs cases as cruelty or abandonment cases, based on the average hourly rate for Inspectors of AUD 35/h, assuming 3 h per week for the period of one year in total are spent assisting potential RSPCA NSW Community Programs cases, 50% of which are likely to be related to the Emergency Boarding program.	RSPCA Inspectors frequently described their frustration at having to case-manage individuals who would be more appropriately managed through RSPCA NSW Community Programs and that this takes up a considerable amount of their time at work. Hence, we use the value to the Inspectors of being able to use this time for pursuing genuine cruelty offences as a proxy for this outcome.	AUD 2730	Inspector interviews
Improved mental health	Observed spending on related goods. The cost of a typical mental health plan of six sessions with a psychologist at AUD 210 per session.	RSPCA NSW Inspectors described being relieved of substantial moral distress when they can refer clients to the RSPCA NSW Community Programs. Without this referral pathway, Inspectors’ mental health can be negatively affected. We consider this moral distress comparable with mild anxiety and hence value this outcome using a typical treatment plan for mild anxiety.	AUD 1260	Australian Psychological Society [23]

**Table 9 animals-13-02264-t009:** Valuing outcomes experienced by animal pounds and shelters because of the RSPCA NSW Emergency Boarding and Homelessness program.

Outcome	Proxy Description	Rationale	Value	Source
Fewer animals abandoned without care	Observed spending on related goods. Cost to RSPCA of processing an abandoned animal	According to interviews with RSPCA Inspectors, animals regularly come into the care of the RSPCA shelter via the Inspectorate because of being abandoned without care. This was also mentioned in interviews with Emergency Boarding program clients. Costs are incurred by RSPCA NSW for retrieving, sheltering, and rehabilitating these animals. Hence, the average costs associated with rehabilitating a typical Emergency Boarding program client’s animal (a medium sized adult dog) has been used as the proxy for this outcome.	AUD 885	RSPCA records
Fewer animals surrendered by their owner	Observed spending on related goods. Cost to RSPCA of processing a surrendered animal	When asked ‘What do you think would be different for you now if you had not accessed assistance for your animal/s through RSPCA?’ some Emergency Boarding program clients responded that their animal would have been surrendered to a pound or shelter. Hence, the average cost to RSPCA NSW of processing a surrendered animal from the time of surrender to adoption has been used as a proxy for this outcome.	AUD 686	RSPCA records

**Table 10 animals-13-02264-t010:** Net social value of outcomes experienced by stakeholders of the RSPCA NSW Emergency Boarding and Homelessness program.

**Stakeholder** **N ^1^**	**Outcome**	**Financial Proxy**	**Outcome Incidence**	**Deadweight**	**Attribution**	**Displacement**	**Benefit Period** **(Years)**	**Drop-Off**	**Net Social Value ^2^**
Clients 259	Extended or enhanced human–animal bond	$3453	93%	7%	0%	25%	8	5%	$1,554,223
	Improved mental health and wellbeing	$5320	90%	7%	50%	0%	1	N/A	$574,429
	Decreased financial stress	$2058	76%	7%	0%	0%	1	N/A	$376,068
	Increased social inclusion/decreased isolation	$2778	72%	7%	50%	0%	1	N/A	$242,274
	Improved personal safety	$3427	41%	7%	75%	0%	1	N/A	$85,394
	Improved access to care for themselves	$413	59%	7%	50%	0%	1	N/A	$29,143
	Improved physical health	$1140	38%	7%	75%	0%	1	N/A	$26,039
Client’s animal/s 627	Improved wellbeing as a result of preserving or improving the human–animal bond	$1380	91%	25%	0%	25%	8	0%	$1,249,542
	Access to safe accommodation	$1925	71%	7%	0%	0%	1	N/A	$801,776
	Improved physical health	$241	71%	7%	0%	0%	1	N/A	$100,315
RSPCA Inspectors 35	More time available to pursue genuine animal cruelty offenses	$2730	80%	7%	0%	0%	1	N/A	$71,089
	Improved mental health	$1260	80%	7%	0%	0%	1	N/A	$32,810
RSPCA NS shelters	Fewer animals abandoned without care (54 animals; 9% of clients reported their animal would have been abandoned without care)	$885	100%	7%	0%	0%	1	N/A	$44,246
	Fewer animals surrendered by their owner (233 animals; 37% of clients reported their animal would have been rehomed or surrendered)	$686	100%	25%	25%	0%	1	N/A	$89,832

^1^ The total number of people/animals in each stakeholder group during the 2020–2021 financial year. ^2^ To be conservative when claiming, the net present value of outcomes with a benefit period of nine years has only included the value of these outcomes for the first three years.

**Table 11 animals-13-02264-t011:** Net social value of outcomes experienced by stakeholders of the RSPCA NSW Emergency Boarding and Homelessness program.

Value created for program clients	$2,887,569
Value created for client’s animals	$2,151,633
Value created for RSPCA NSW Inspectors	$103,900
Value created for animal pounds and shelters	$134,078
Total social value created for all stakeholders	$5,277,179
Net Program Investment	$642,489
**Social return for each $1 invested**	**$8.21**

## Data Availability

Data supporting reported results can be found at [LINK TBA].

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
