# Peer review of "Emergency Animal Boarding: A Social Return on Investment"

_animals, 2023, doi:10.3390/ani13142264_

Round 1
Reviewer 1 Report
This ms describes a cost-benefit analysis of an emergency animal housing program. It is a very useful contribution to the literature, especially since the authors managed to figure out how to cost intangibles like mental health and the human-animal bond. Before recommending it for publication, I have a few suggestions to improve clarity .
My major criticism is that part of the Results would be better in the Methods, and some of the Methods are not sufficiently described for someone else to replicate the study. See specific comments below.
Intro
Generally good. Clearly structured and well-written.
L46 - please cite the claim about the times when the HAB is of most value
L59 - please cite the claim about animals living with homeless people having 'excellent' welfare. Surely there are risks associated with constant outdoor living that must also be noted? All that freedom comes at the cost of potentially being run over, attacked by other animals or humans, and disease
Methods
Section 2.4 - what is the Theory of Change workshop? This is not described in the intro so it's not clear what is meant here. Please expand so that anyone replicating the study will have sufficient info to run their own Theory of Change workshop.
Section 2.5 - please describe the qualitative analysis in more detail. Were the interviews transcribed? If so, by whom? Who did the initial analysis? Was there a second coder? What sort of software was used to categorise the findings? if not, how did the authors determine validity and reliability of the findings? Also, mention that the survey was analysed for frequencies.
Please briefly explain deadweight, displacement, benefit period, and drop-off. They are already described in the Results, but they should be explained here as well. The explanation belongs in the methods.
How did the authors determine the financial proxies for each social value? Again, I know these are explained in the results, but they should really be described here. They are part of the methods, not the results.
Results
L199 - I thought that the theory of change workshop took place before the interviews? Here it sounds like the interviews were used to establish the theory of change presented in the workshop. Providing more info about these workshops in the methods may help make this clearer.
L203-205, plus Table 1, belong in the methods.
L266-271 - this para, along with others later on (e.g., L292-296) should be clearly marked as based on the survey, rather than the interviews. Right now they are getting jumbled a bit, and it sounds like the authors are trying to put qualitative findings into a quant format. Suggest just adding a brief mention of the survey in these places to make it very clear to the reader. Also, for both of these segments, the results are the same as those presented in the tables, so either the text can be dramatically shortened to be purely interpretive, or the tables can go. There's no need to present the full results in both formats.
Table 1 - suggest adding 'based on survey responses' to the end of the caption for clarity's sake
For Tables 1-4 - suggest adding Ns in addition to % in the tables.
L344-348 - why are there two client quotes here, rather than quotes from animal shelter employees? The section is focused on the findings from the animal shelter group so it doesn't make sense to have client quotes here.
L356 - please briefly describe the negative experience, in such a way that there is no risk of the participant being identified. It is difficult to imagine what a negative experience might look like in this context, so more info would be useful. It may also help orgs who are thinking of setting up a program like this, work to avoid those negative experiences.
Table 5 - good information in this table, but the formatting makes it difficult to understand what content is on which row. This is probably to do with the template, but it could be improved. For instance, having a landscape orientation may save space because the 'rationale' column can be lengthened out. Also, having spaces between the rows would also help make it clearer. If the text can be formatted to start at the top of the row, rather than in the centre, that would also help, if the Animals team will go for it.
Table 5 - 'extended or enhanced HAB' row - is the USD 2400 figure per year or for 8 years?
Table 5 - 'increased social inclusion' - why the 6 months for recreation? That is not well-justified, unlike the rest of the info in this table.
Table 7 - 'more time available to pursue genuine animal cruelty' - is the value per year?
Table 9 - explain that the N is not the number of participants, but the total number of beneficiaries/employees/animals in RSPCA care during the relevant period.
L487 - total input was $642K - is that the total for all 259 people/627 animals/35 inspectors or just the participants in the study?
Discussion
The discussion is good but very 'big picture'. There needs to be more about the study itself, including its advantages and limitations, and future directions. Please add a short para with that info or embed it throughout the existing paras.
L569 - '...hospitalised for physically or mentally...' needs 'being' in front of 'physically'
Conflict of interest statement - it would be useful to understand how the authors' employment at RSPCA NSW might have influenced the results. Were any steps taken to avoid a potential bias? If so, please describe. This will strengthen the piece by proactively answering questions that may arise in the minds of the readers.
Appendix A L615 what is meant by 'stated preference'?
Appendix B L643 - was this question a multiple response option or did they have to select the one most appropriate to their situation?
Appendix B - it would have been good to ask the participants how the RSPCA could have done better. Please add this as a limitation.
The English is generally fine but in some places a word or two is missing or extraneous, or the tense is incorrect (e.g., L85 'what changes' should just be 'changes'; L128-130 should be past tense, not present).
Reviewer 2 Report
This is an interesting and valuable research project. It would benefit from some discussion of the need for such an analysis beyond an interest in evaluating the impact and value of the Emergency Boarding Program. Have issues been raised about the worth of the program? Have there been criticisms that the goals deviate too far from conventional SPCA activities? Such questions do arise in the US when humane organizations extend their reach beyond basic animal protection and care.
Given that the primary focus of the research is to determine the cost/benefits of the program - a key element is the determination of financial proxies for the presumed benefits. This process needs to be described in more detail than simply stating (line 168) that “desktop research was undertaken”. Were all the authors involved in the choice of proxies? Was there consensus of what proxies to use? Was there any consultation with outside experts as to potential appropriate proxies? Were any proxy ideas discarded? It might have been useful to involve the non-client participants to suggest possible proxies for the benefits they identified.
The selection of proxies seems to be potentially problematic, although the authors do a good job of attempting to document the rationale behind each proxy choice. Despite this, several of the proxies seem to be a stretch or even dubious. Specifically using average amount spent on recreation for six months as a proxy for decreased social isolation seems questionable. many elements considered “recreation” might not involve social inclusion - e.g. attending a movie or dining alone! Likewise using cost of a gym membership as a proxy for improved physical health seems weak. A better measure might involve costs of medical visits avoided due to improved health or self-care. Using pet-insurance as a proxy for improved well-being also seems a stretch. A better measure might be costs of urgent or emergency care avoided through better bonding and pet-care.
Even without these potentially questionable proxies, the key finding of significant benefits over costs would still remain and the overall significance of the research would not be impacted.
I was disappointed that virtually no attention was given to a major source of demand for emergency animal boarding - providing for pets of victims escaping from domestic violence. Currently there are more than 300 programs in the US structured to provide pet and people friendly sheltering in situations of interpersonal violence, likely more than the number of programs addressing pets and homelessness. The paper mentions that RSPCA NSW has a Domestic Violence Program and it is likely that a similar analysis would identify an even stronger social return on investment in these cases. I am aware of at least five organizations in NSW participating in such programs:
Vinnies Women’s Crisis Centre (Adelaide, NSW Australia)
Dignity (Northmead, NSW Australia) 1300 332 334
Domestic Violence Service Management (Sydney, NSW, Australia) 02-9251-2405
Jessie Street Domestic Violence Services, Inc. (NSW, Australia) 02-9622-7999
Tamworth Family Support Services (Tamworth, NSW, Australia) 02-6763-2333
The omission of this population does not distract from the value of the current manuscript, but I would encourage the authors to conduct a similar analysis for a future publication.
